# Evaluation of the impact of COVID-19 pandemic on hospital admission related to common infections: Risk prediction models to tackle antimicrobial resistance in primary care

Ali Fahmi[1]*, Victoria Palin[1,2], Xiaomin Zhong[1], Ya-Ting Yang[1], Simon Watts[3], Darren M. Ashcroft[4,5], Ben Goldacre[6], Brian MacKenna[6], Louis Fisher[6], Jon Massey[6], Amir Mehrkar[6], Seb Bacon[6], the OpenSAFELY Collaborative[6], Kieran Hand[7], Tjeerd Pieter van Staa[1]

1 Centre for Health Informatics, Faculty of Biology, Medicine and Health, School of Health Sciences, the University of Manchester, Manchester, United Kingdom, 2 Maternal and Fetal Health Research Centre, Division of Developmental Biology and Medicine, the University of Manchester, Manchester, United Kingdom, 3 North West Ambulance Service, Manchester NHS Foundation Trust, Manchester, United Kingdom, 4 Centre for Pharmacoepidemiology and Drug Safety, Faculty of Biology, Medicine and Health, School of Health Sciences, University of Manchester, Manchester, United Kingdom, 5 NIHR Greater Manchester Patient Safety Translational Research Centre, Faculty of Biology, Medicine and Health, School of Health Sciences, University of Manchester, Manchester, United Kingdom, 6 Nuffield Department of Primary Care Health Sciences, Bennett Institute for Applied Data Science, University of Oxford, Oxford, United Kingdom, 7 NHS England, London, United Kingdom

* ali.fahmi@manchester.ac.uk

## Abstract

### Background

Antimicrobial resistance (AMR) is a multifaceted global challenge, partly driven by inappropriate antibiotic prescribing. The objectives of this study were to evaluate the impact of the COVID-19 pandemic on treatment of common infections, develop risk prediction models and examine the effects of antibiotics on infection-related hospital admissions.

### Methods

With the approval of NHS England, we accessed electronic health records from The Phoenix Partnership (TPP) through OpenSAFELY platform. We included adult patients with primary care diagnosis of common infections, including lower respiratory tract infection (LRTI), upper respiratory tract infections (URTI), and lower urinary tract infection (UTI), from 1 January 2019 to 31 August 2022. We excluded patients with a COVID-19 record in the 90 days before to 30 days after the infection diagnosis. Risk prediction models using Cox proportional-hazard regression were developed for infection-related hospital admission in the 30 days after the common infection diagnosis.

### Results

We found 12,745,165 infection diagnoses from 1 January 2019 to 31 August 2022. Of them, 80,395 (2.05%) cases were admitted to the hospital during follow-up. Counts of hospital

**Data Availability Statement:** "All data were linked, stored and analysed securely within the

OpenSAFELY platform https://opensafely.org/. Data include pseudonymized data such as coded diagnoses, medications and physiological parameters. No free text data are included. All code is shared openly for review and re-use under MIT open license (https://github.com/opensafely/amr-uom-brit/). Detailed pseudonymised patient data is potentially re-identifiable and therefore not shared. We rapidly delivered the OpenSAFELY data analysis platform without prior funding to deliver timely analyses on urgent research questions in the context of the global COVID-19 health emergency: now that the platform is established, we are developing a formal process for external users to request access in collaboration with NHS England; details of this process will be published shortly on OpenSAFELY.org."

**Funding:** This work was supported by Health Data Research UK (Better prescribing in frail elderly people with polypharmacy: learning from practice and nudging prescribers into better practice -BetterRx) and by National Institute for Health and Care Research (NIHR130581 - Cluster randomised trial to improve antibiotic prescribing in primary care: individualised knowledge support during consultation for general practitioners and patients – BRIT2). DMA is funded by the NIHR Greater Manchester Patient Safety Translational Research Centre (PSTRC-2016-003). The views expressed are those of the authors and not necessarily those of Health Data Research UK, the NHS, the NIHR, the Department of Health and Social Care or Public Health England. AF is funded by the National Institute for Health and Care Research (DSE Award; NIHR303781). The funders had no role in study design, data collection and analysis, decision to publish, or preparation of the manuscript.

**Competing interests:** All authors declare the following: BG and OpenSAFELY has received research funding from the Laura and John Arnold Foundation, the NHS National Institute for Health Research (NIHR), the NIHR School of Primary Care Research, NHS England, the NIHR Oxford Biomedical Research Centre, the Mohn-Westlake Foundation, NIHR Applied Research Collaboration Oxford and Thames Valley, the Wellcome Trust, the Good Thinking Foundation, Health Data Research UK, the Health Foundation, the World Health Organisation, UKRI MRC, Asthma UK, the British Lung Foundation, and the Longitudinal Health and Wellbeing strand of the National Core Studies programme; he is a Non-Executive Director at NHS Digital; he also receives personal income from speaking and writing for lay audiences on the misuse of science. AM has received consultancy fees (from https://inductionhealthcare.com) and is

admission for infections dropped during COVID-19, for example LRTI from 3,950 in December 2019 to 520 in April 2020. Comparing those prescribed an antibiotic to those without, reduction in risk of hospital admission were largest with LRTI (adjusted hazard ratio (aHR) of 0.35; 95% confidence interval (CI), 0.35–0.36) and UTI (aHR 0.45; 95% CI, 0.44–0.46), compared to URTI (aHR 1.04; 95% CI, 1.03–1.06).

## Conclusions

A substantial variation in hospital admission risks between infections and patient groups was found. Antibiotics appeared more effective in preventing infection-related complications with LRTI and UTI, but not URTI. While this study has several limitations, the results indicate that a focus on risk-based antibiotic prescribing could help tackle AMR in primary care.

## Introduction

Antimicrobial resistance (AMR) is a multifaceted global challenge that needs to be managed through antimicrobial stewardship interventions [1, 2]. Antibiotics are prescribed to prevent infections, but if prescribed inappropriately or excessively, antibiotics use can drive AMR [3]. Prescribing of antibiotics declined between the end of 2019 and 2021 compared to previous years as an indirect impact of the COVID-19 pandemic [2], mainly due to reduced social mixing and spread of infections.

Few studies have evaluated the risk of hospital admissions related to common infections and antibiotic prescribing during the COVID-19 pandemic. During the pandemic, Zhu et al. found a reduction in community antibiotics prescribing in northwest London [4]. Silva et al. evaluated the impact of the pandemic on the trend of antibiotics prescribing in outpatient care in Portugal and found a significant reduction in antibiotic prescribing in outpatient care [5]. Several pre-pandemic studies have investigated the link between using antibiotics and developing complications, for example Mistry et al. evaluated the risk of incident complication related to urinary tract infection (UTI), upper respiratory tract infection (URTI), and lower respiratory tract infection (LRTI) [3]. van Bodegraven et al. found an association between lower rates of antibiotics prescribing and higher risk of infection-related complications [6]. Whilst these studies are informative, there is a need to understand the impact of the pandemic on outcomes after common infections. This study aimed to evaluate the impact of the COVID-19 pandemic on the primary care treatment with antibiotic for common infections in England and to develop and validate risk prediction models for infection-related complications. Risk prediction models are statistical models that aim to predict the probability of future events, for example whether a patient will develop a disease or not. This study was part of the BRIT2 project that aims to optimise the use of antibiotics for treatment of common infections in primary care [7].

## Methods

We used data from the OpenSAFELY platform (https://opensafely.org/) that focuses on urgent research into COVID-19 pandemic and securely links, pseudonymises, stores, and analyses I on behalf of the National Health Service pandemic [8, 9]. This platform provides almost 24 million people's pseudonymised patient-level data of primary care from The Phoenix Partnership (TPP). These data are linked to external databases, e.g., Hospital Episode Statistics with

member of RCGP health informatics group and the NHS Digital GP data Professional Advisory Group that advises on access to GP Data for Pandemic Planning and Research (GDPPR). For the latter, he received payment for the GDPPR role. This does not alter our adherence to PLOS ONE policies on sharing data and materials. There are restrictions on access to the data of this study since the data are patient-level and potentially re-identifiable, as described in the Data access and verification section.

**Abbreviations:** AMR, Antimicrobial Resistance; UTI, Urinary Tract Infection; URTI, Upper Respiratory Tract Infection; LRTI, Lower Respiratory Tract Infection; TPP, The Phoenix Partnership; BMI, Body Mass Index; EHR, Electronic Health Record; ICD-10, 10th revision of International Classification of Diseases; SGSS, Second Generation Surveillance System; GP, General Practitioner; IMD, Index of Multiple Deprivation; CCI, Charlson Comorbidity Index; aHR, adjusted Hazard Ratio; cHR, crude Hazard Ratio; UK, United Kingdom; NICE, National Institute for Health and Care Excellence.

hospital admission diagnoses [10]. The OpenSAFELY data include patient-level demographics (age, sex, Body Mass Index or BMI, ethnicity, and smoking status), clinical diagnoses history, medication history, and vaccination history. OpenSAFELY's approved projects need to obtain ethics approval and then develop the codes and maintain them in a public GitHub repository and use OpenSAFELY Jobs interface to run the codes against real data (e.g., GitHub repository of BRIT2 project: https://github.com/opensafely/amr-uom-brit) [9]. Our study protocol was approved by the Research Ethics Committee on 17 August 2021 (reference 21/SC/0287). Patients' consent was not required since the data were anonymised electronic health records (EHRs) and for retrospective research use. The start date of data access and data analysis was on 8 September 2021 (as indicated by the *created_at* parameter of GitHub API for BRIT2 project repository: https://api.github.com/repos/opensafely/amr-uom-brit).

## Study population

The inclusion criteria were adult individuals registered with a general practice during study period (1 January 2019 to 31 August 2022), who had diagnosis of a common infection, namely LRTI, URTI (including specific URTI, cough, cold with cough, and sore throat), lower UTI (not including renal infections), sinusitis, otitis media, and otitis externa. The reason for restricting to adults was that the prevalence of common infections is different in children and that the BRIT2 project is focusing on adults [7]. Individuals with a record of COVID-19 diagnosis in the 90 days before or 30 days after the date of infection diagnosis were excluded. This was based on data from the Second Generation Surveillance System (SGSS) with positive COVID-19 tests and those of the general practitioners (GPs) with a COVID-19 diagnosis. The diagram of study design shows the inclusion criterion of any infection-related hospital admission and the exclusion criterion of having any COVID-19 diagnosis records (Fig 1).

The outcomes of interest were infection-related hospital admissions in the 30-day after the date of infection diagnosis (i.e., follow-up period). We used the 10th revision of

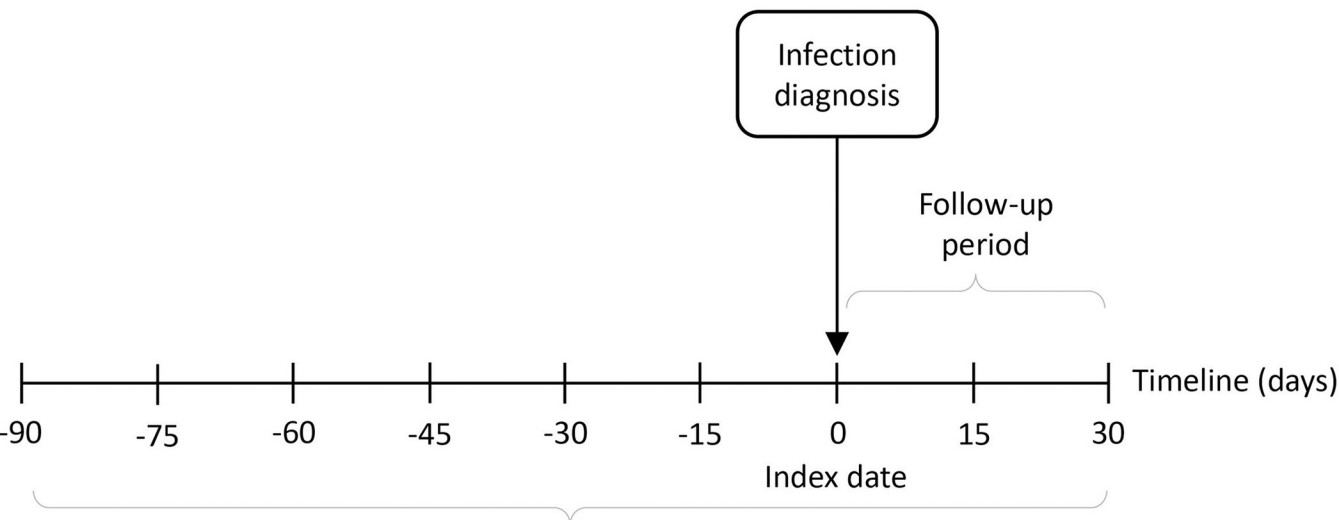

**Fig 1. Diagram of study design.**

International Classification of Diseases (ICD-10) as recorded in the primary admission diagnosis of the linked hospital data. The selected ICD-10 codes are available at opencode-lists website [11]. The dictionary of medicines and devices (DMD) codes were used to identify antibiotics [12].

### Study variables

We extracted predictor variables that may be associated with infection-related hospital admissions. These included age, sex, ethnicity, smoking status, socioeconomic class using the Index of Multiple Deprivation (IMD), region, BMI, comorbidities measured with the Charlson Comorbidity Index (CCI) [13], season of infection diagnosis, flu vaccination in the one year before, and history of prior antibiotics in the one year before.

### Statistical methods

The common infection cohorts were split into four sub-datasets, including incident infections with and without prescribed antibiotics and prevalent infections with and without prescribed antibiotics. Incident common infections were those without a diagnosis record for the same infection in the 42 days before. Prescribing of antibiotics was based on those given on the date of infection records or in the five days after the infection diagnosis. We analysed the count and rate of infection-related hospital admissions for each sub-dataset. A Cox proportional-hazards regression model was fitted to each sub-dataset. Censored patients were those who died or deregistered from the practice during follow-up. Patients with a missing value for ethnicity, smoking status, IMD, and BMI variables were given a missingness indicator.

Each sub-dataset for each common infection was randomly split into development (75%) and validation (25%) cohort. To assess the effect of prescribed antibiotics on hospital admission, Cox models were fitted with an additional predictor binary variable for prescribed antibiotics. We also built Cox models with an additional categorical variable of antibiotic type (most prescribed, second most prescribed, others, and none). For evaluation of impact of the COVID-19 pandemic, we analysed the count of infection-related hospital admission considering pre-pandemic, beginning of the pandemic, pandemic and after the second national lockdown. The performance of Cox models was evaluated with C-statistics, which measures the models' ability to discriminate patients with complication and those without. Calibration of models was calculated by comparing the observed and predicted risks for deciles of predicted risks of infection-related hospital admission. The Jupyter notebooks are available at https://github.com/opensafely/amr-uom-brit/tree/hosp_pred. The Python lifelines package version 0.26.4 was used for the Cox models [14].

## Results

### Baseline characteristics

A total of 12,745,165 diagnoses of common infections was found from 1 January 2019 to 31 August 2022, of which 11,455,025 (89.88%) were incident and 1,290,140 (10.12%) were prevalent. Of incident common infections, 7,539,015 (65.81%) were prescribed antibiotics (86.33% received antibiotics for incident LRTI, 55.93% for URTI, and 86.79% for UTI). Of prevalent common infections, 789,340 (61.18%) were prescribed antibiotics. The main baseline characteristics of the cohort of incident common infections (namely LRTI, URTI, and UTI) without prescribed antibiotics are shown (Table 1). Remaining baseline characteristics of these cohorts as well as those for other infections are presented in S1-S4 Tables in S1 Appendix.

**Table 1. Baseline characteristics of the cohorts of incident common infections in patients not prescribed antibiotics (using data from 1 January 2019 to 31 August 2022).**

| | LRTI[1] | URTI[2] | UTI[3] |
|---|---|---|---|
| **Total, N infections** | 252,975 | 2,567,235 | 294,325 |
| **Age, N (%)** | | | |
| 18–24 | 10,150 (4.01) | 158,300 (6.17) | 25,545 (8.68) |
| 25–34 | 20,185 (7.98) | 263,190 (10.25) | 35,195 (11.96) |
| 35–44 | 22,390 (8.85) | 267,840 (10.43) | 28,180 (9.57) |
| 45–54 | 31,215 (12.34) | 372,565 (14.51) | 33,920 (11.52) |
| 55–64 | 39,005 (15.42) | 480,850 (18.73) | 36,890 (12.53) |
| 65–74 | 50,195 (19.84) | 548,215 (21.35) | 48,330 (16.42) |
| 75+ | 79,830 (31.56) | 476,275 (18.55) | 86,270 (29.31) |
| **Sex, N (%)** | | | |
| Male | 108,850 (43.03) | 1,116,735 (43.50) | 83,720 (28.45) |
| Female | 144,125 (56.97) | 1,450,500 (56.50) | 210,605 (71.55) |
| **Ethnicity, N (%)** | | | |
| White | 145,930 (57.69) | 1,519,395 (59.18) | 168,765 (57.34) |
| Non-White | 15,905 (6.29) | 190,065 (7.40) | 19,250 (6.54) |
| Unknown | 91,135 (36.03) | 857,780 (33.41) | 106,310 (36.12) |
| **CCI[4], N (%)** | | | |
| Very low (= 0) | 123,435 (48.79) | 1,499,380 (58.40) | 176,150 (59.85) |
| Low (= 1) | 95,705 (37.83) | 846,450 (32.97) | 85,315 (28.99) |
| Medium (= 2) | 26,145 (10.33) | 179,440 (6.99) | 25,080 (8.52) |
| High (= 3 and = 4) | 5,645 (2.23) | 32,810 (1.28) | 5,900 (2.00) |
| Very high ($\geq 5$) | 2,040 (0.81) | 9,160 (0.36) | 1,875 (0.64) |
| **Flu vaccination, N (%)** | | | |
| Yes | 134,495 (53.17) | 1,237,915 (48.22) | 133,530 (45.37) |
| No | 118,475 (46.83) | 1,329,320 (51.78) | 160,800 (54.63) |
| **Period** | | | |
| Pre-pandemic | 103,175 (40.78) | 990,915 (38.60) | 91,650 (31.14) |
| Beginning and during pandemic | 67,095 (26.52) | 723,000 (28.16) | 91,120 (30.96) |
| After 2nd lockdown | 82,705 (32.69) | 853,320 (33.24) | 111,555 (37.90) |
| **Count of antibiotic prescription in the one year before, mean (SD[5])** | 2.20 (2.86) | 1.42 (2.09) | 2.30 (3.09) |

[1] LRTI, Lower Respiratory Tract Infection.

[2] URTI, Upper Respiratory Tract Infection.

[3] UTI, Urinary Tract Infection.

[4] CCI, Charlson Comorbidities Index, measured from 17 weighted conditions, including myocardial infarction, congestive heart failure, peripheral vascular disease, cerebrovascular disease, dementia, chronic pulmonary disease, Connective tissue disease, ulcer disease, mild liver disease, diabetes, hemiplegia, moderate or severe renal disease, diabetes with complications, any malignancy (including leukaemia and lymphoma), moderate or severe liver disease, metastatic solid tumour, and AIDS.

[5] SD, standard deviation.

## Counts and rates of infection-related hospital admissions

The counts and rates of infection-related hospital admissions are presented (Table 2 and S5-S8 Tables in S2 Appendix). There were 268,805 cases of infection-related hospital admission within 30-day follow-up after infection diagnosis. Rates were highest in patients with the highest CCI (rate of 149.5 in LRTI, 58.4 in URTI, and 168.0 in UTI). Although the counts of LRTI

**Table 2. Count and rate of hospital admission related to incident common infections in patients not prescribed antibiotics (using data from 1 January 2019 to 31 August 2022).**

|  | LRTI[1] | URTI[2] | UTI[3] |
|---|---|---|---|
| **Total, N cases** | 17,915 | 40,035 | 18,140 |
| **Age, N (rate[4])** |  |  |  |
| 18–24 | 235 (23.2) | 1,560 (9.9) | 455 (17.8) |
| 25–34 | 505 (25.0) | 2,150 (8.2) | 615 (17.5) |
| 35–44 | 670 (29.9) | 2,150 (8.0) | 565 (20.0) |
| 45–54 | 1,210 (38.8) | 3,180 (8.5) | 1,000 (29.5) |
| 55–64 | 2,030 (52.0) | 4,965 (10.3) | 1,545 (41.9) |
| 65–74 | 3,870 (77.1) | 8,495 (15.5) | 3,525 (72.9) |
| 75+ | 9,395 (117.7) | 17,540 (36.8) | 10,435 (121.0) |
| **Sex, N (rate[4])** |  |  |  |
| Male | 8,690 (79.8) | 19,235 (17.2) | 8,160 (97.5) |
| Female | 9,225 (64.0) | 20,800 (14.3) | 9,980 (47.4) |
| **Ethnicity, N (rate[4])** |  |  |  |
| White | 10,255 (70.3) | 22,810 (15.0) | 10,310 (61.1) |
| Non-White | 780 (49.0) | 2,120 (11.2) | 705 (36.6) |
| Unknown | 6,880 (75.5) | 15,105 (17.6) | 7,125 (67.0) |
| **CCI[5], N (rate[4])** |  |  |  |
| Very low (= 0) | 6,520 (52.8) | 16,495 (11.0) | 7,210 (40.9) |
| Low (= 1) | 7,400 (77.3) | 15,580 (18.4) | 6,790 (79.6) |
| Medium (= 2) | 2,940 (112.4) | 5,845 (32.6) | 2,940 (117.2) |
| High (= 3 and = 4) | 750 (132.9) | 1,580 (48.2) | 880 (149.2) |
| Very high (≥5) | 305 (149.5) | 535 (58.4) | 315 (168.0) |
| **Flu vaccination, N (rate[4])** |  |  |  |
| Yes | 11,765 (87.5) | 24,535 (19.8) | 11,660 (87.3) |
| No | 6,150 (51.9) | 15,500 (11.7) | 6,475 (40.3) |
| **Periods, N (rate[4])** |  |  |  |
| Pre-pandemic | 7,375 (71.5) | 16,165 (16.3) | 6,125 (66.8) |
| Beginning and during pandemic | 4,575 (68.2) | 10,440 (14.4) | 5,170 (56.7) |
| After 2nd lockdown | 5,965 (72.1) | 13,430 (15.7) | 6,845 (61.4) |

[1] LRTI, Lower Respiratory Tract Infection.

[2] URTI, Upper Respiratory Tract Infection.

[3] UTI, Urinary Tract Infection.

[4] Rate is the number of cases per 1000 patients with common infection, calculated by dividing the count of infection-related hospital admission cases (numerator) by the count of infection diagnosis (denominator) and then multiplied by 1000.

[5] CCI, Charlson Comorbidities Index, measured from 17 weighted conditions, including myocardial infarction, congestive heart failure, peripheral vascular disease, cerebrovascular disease, dementia, chronic pulmonary disease, Connective tissue disease, ulcer disease, mild liver disease, diabetes, hemiplegia, moderate or severe renal disease, diabetes with complications, any malignancy (including leukaemia and lymphoma), moderate or severe liver disease, metastatic solid tumour, and AIDS.

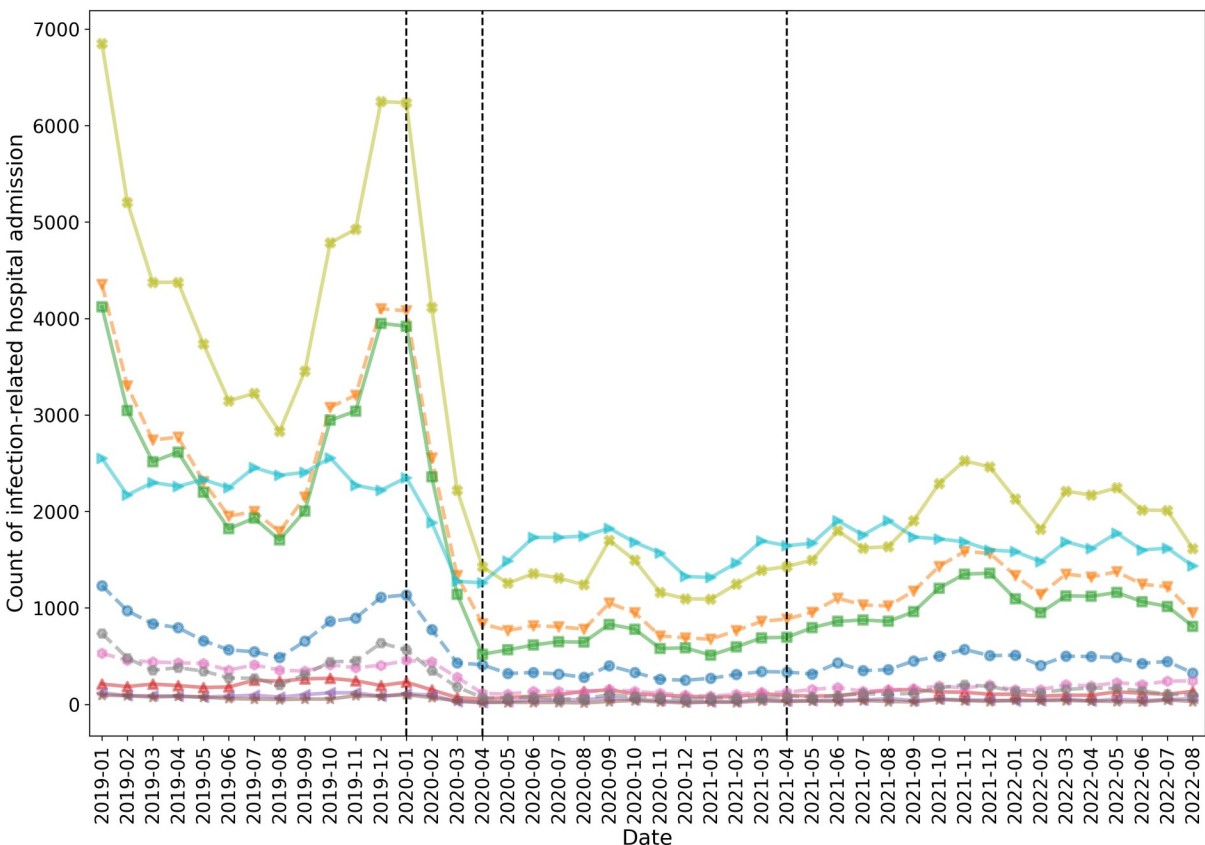

**Fig 2. Count of hospital admissions related to upper respiratory tract infections (URTI; including specific URTI, cough, cold with cough, and sore throat), lower respiratory tract infection (LRTI), urinary tract infection (UTI), sinusitis, otitis media, and otitis externa.**

and URTI dropped during the pandemic, the rates did not change substantially by COVID-19 status (Table 2). Rates of UTI, however, fluctuated; from 66.8 pre-pandemic to 56.7 in the beginning and during the pandemic periods, and then to 61.4 after the second lockdown. A reduction of infection-related hospital admission was revealed in the beginning of the pandemic, between January 2020 and April 2020, especially in LRTI and URTI which dropped by 87% and 77%, respectively, between January 2020 and April 2020 (Fig 2).

## Performance and hazard ratio results

C-statistics of the Cox models with development and validation datasets of incident infections with no antibiotics were respectively 0.68 and 0.67 for LRTI, 0.73 and 0.72 for URTI, and 0.73 and 0.73 for UTI (S9 Table in S3 Appendix). C-statistics of the Cox models with development and validation datasets for hospital admission related to all incident and prevalent infections with and without antibiotics are presented in the same table (S9 Table in S3 Appendix). Infections included LRTI, URTI, UTI, sinusitis, otitis media, otitis externa, as well as the components of the URTI: URTI, cough, cold with cough, and sore throat, with and without antibiotics.

Age was strongly associated with infection-related complications with adjusted hazard ratios (aHRs) of age category 75+ being 5.27 (95% CI 4.49 to 6.28) in LRTI, 3.54 (95% CI 3.31 to 3.78) in URTI, and 5.54 (95% CI 4.94 to 6.20) in UTI (Table 3). CCI also had a high association with hospital admission related to incident infections as aHRs of very high CCI were 2.03

**Table 3. Adjusted hazard ratios of infection-related hospital admissions for several predictors for incident common infection stratified by infection (using data from 1 January 2019 to 31 August 2022).**

| | LRTI[1] | URTI[2] | UTI[3] |
|---|---|---|---|
| | Adjusted HR[4] (95% CI[5]) | Adjusted HR[4] (95% CI[5]) | Adjusted HR[4] (95% CI[5]) |
| **Sex** | | | |
| Male | 1.22 (1.18–1.26) | 1.18 (1.16–1.21) | 1.54 (1.49–1.59) |
| **Age** | | | |
| 25–34 | 1.15 (0.96–1.39) | 0.90 (0.83–0.97) | 0.99 (0.86–1.13) |
| 35–44 | 1.47 (1.23–1.75) | 0.91 (0.84–0.98) | 1.04 (0.90–1.20) |
| 45–54 | 1.78 (1.50–2.10) | 0.96 (0.89–1.03) | 1.43 (1.26–1.63) |
| 55–64 | 2.38 (2.02–2.80) | 1.13 (1.05–1.21) | 1.94 (1.72–2.19) |
| 65–74 | 3.36 (2.86–3.95) | 1.67 (1.56–1.78) | 3.27 (2.92–3.68) |
| 75+ | 5.27 (4.49–6.18) | 3.54 (3.31–3.78) | 5.54 (4.94–6.20) |
| **BMI[6]** | | | |
| Underweight | 1.11 (1.01–1.22) | 1.39 (1.30–1.49) | 1.13 (1.01–1.27) |
| Overweight | 0.94 (0.89–0.98) | 0.85 (0.83–0.88) | 0.93 (0.89–0.97) |
| Obese | 1.04 (0.99–1.09) | 0.99 (0.96–1.02) | 1.13 (1.08–1.19) |
| Unknown | 1.15 (1.09–1.22) | 1.32 (1.27–1.37) | 1.11 (1.05–1.17) |
| **Ethnicity** | | | |
| White | 1.08 (0.99–1.18) | 1.10 (1.04–1.16) | 1.15 (1.04–1.26) |
| Unknown | 1.13 (1.04–1.24) | 1.24 (1.17–1.32) | 1.26 (1.15–1.39) |
| **CCI[7]** | | | |
| Low | 1.14 (1.10–1.19) | 1.31 (1.28–1.35) | 1.28 (1.23–1.33) |
| Medium | 1.44 (1.37–1.52) | 1.82 (1.75–1.89) | 1.54 (1.46–1.62) |
| High | 1.62 (1.48–1.77) | 2.43 (2.28–2.58) | 1.84 (1.70–2.00) |
| Very high | 2.03 (1.78–2.31) | 3.18 (2.88–3.51) | 2.21 (1.94–2.52) |
| **Smoking status** | | | |
| Smoker | 1.14 (1.08–1.21) | 0.91 (0.88–0.94) | 1.21 (1.14–1.29) |
| Never smoked | 0.94 (0.91–0.98) | 1.15 (1.11–1.18) | 0.97 (0.94–1.01) |
| Unknown | 0.87 (0.60–1.26) | 1.15 (0.96–1.39) | 1.06 (0.76–1.47) |
| **IMD[8]** | | | |
| 1 (most deprived) | 1.02 (0.97–1.08) | 1.04 (1.01–1.08) | 1.07 (1.01–1.13) |
| 3 | 0.91 (0.86–0.96) | 0.92 (0.89–0.96) | 0.91 (0.87–0.96) |
| 4 | 0.94 (0.89–1.00) | 0.94 (0.91–0.98) | 0.88 (0.83–0.93) |
| 5 (most affluent) | 0.83 (0.79–0.88) | 0.88 (0.85–0.91) | 0.76 (0.72–0.81) |
| Unknown | 1.16 (1.01–1.32) | 0.98 (0.89–1.07) | 1.07 (0.94–1.22) |
| **Season** | | | |
| Spring | 0.94 (0.90–0.99) | 0.95 (0.92–0.98) | 0.94 (0.89–0.99) |
| Summer | 1.08 (1.03–1.14) | 0.98 (0.94–1.01) | 1.00 (0.95–1.05) |
| Winter | 1.02 (0.97–1.07) | 1.04 (1.00–1.07) | 0.98 (0.93–1.03) |
| **Region** | | | |
| London | 0.81 (0.72–0.91) | 0.97 (0.90–1.04) | 0.88 (0.79–0.98) |
| North East | 1.04 (0.96–1.12) | 1.02 (0.97–1.07) | 1.04 (0.96–1.13) |
| North West | 0.84 (0.78–0.89) | 0.79 (0.76–0.82) | 0.77 (0.72–0.83) |
| West Midlands | 1.07 (0.98–1.17) | 1.17 (1.10–1.24) | 1.14 (1.05–1.25) |
| Yorkshire and The Humber | 1.02 (0.97–1.08) | 0.94 (0.91–0.98) | 1.11 (1.06–1.18) |
| South East | 1.02 (0.94–1.10) | 0.90 (0.86–0.95) | 0.91 (0.85–0.98) |
| East Midlands | 1.13 (1.08–1.19) | 1.08 (1.04–1.12) | 1.05 (1.00–1.11) |
| South West | 0.98 (0.92–1.04) | 0.83 (0.79–0.86) | 0.83 (0.78–0.88) |

*(Continued)*

**Table 3.** (Continued)

| | LRTI[1] | URTI[2] | UTI[3] |
|---|---|---|---|
| | Adjusted HR[4] (95% CI[5]) | Adjusted HR[4] (95% CI[5]) | Adjusted HR[4] (95% CI[5]) |
| **Flu vaccination** | | | |
| Yes | 0.95 (0.92–0.99) | 0.82 (0.80–0.84) | 0.96 (0.92–1.00) |
| **Count of antibiotic prescription in the one year before** | 1.03 (1.03–1.04) | 1.14 (1.13–1.14) | 1.02 (1.02–1.03) |

[1] LRTI, Lower Respiratory Tract Infection.

[2] URTI, Upper Respiratory Tract Infection.

[3] UTI, Urinary Tract Infection.

[4] HR, hazard ratio.

[5] CI, confidence interval.

[6] BMI, Body Mass Index recorded in the last 5 years.

[7] CCI, Charlson Comorbidities Index, measured from 17 weighted conditions, including myocardial infarction, congestive heart failure, peripheral vascular disease, cerebrovascular disease, dementia, chronic pulmonary disease, Connective tissue disease, ulcer disease, mild liver disease, diabetes, hemiplegia, moderate or severe renal disease, diabetes with complications, any malignancy (including leukaemia and lymphoma), moderate or severe liver disease, metastatic solid tumour, and AIDS.

[8] IMD, Multiple Deprivation Index, quintile measured from patient-level address.

Reference group for variable sex is female, for age is 18–25, for BMI is healthy weight, for ethnicity is non-white, for CCI is very low, for smoking status is ex-smoker, for IMD is 2, for season is autumn, for region is east, for flu vaccination is no.

(95% CI 1.78 to 2.31) for LRTI, 3.18 (95% CI 2.88 to 3.51) for URTI, and 2.21 (95% CI 1.94 to 2.52) for UTI. History of prior antibiotics influenced the risk of infection-related complication, especially in mild infections like URTI with aHR of 1.14 (95% CI 1.13–1.14). aHRs for these predictors for other infections are shown in S10-S12 Tables in S3 Appendix.

Crude hazard ratios (cHRs) of antibiotic exposure (compared to non-exposure) for incident infections were 0.35 (95% CI 0.35 to 0.36) for LRTI, 1.04 (95% CI 1.03 to 1.06) for URTI, and 0.45 (95% CI 0.44 to 0.46) for UTI (Table 4). Patients prescribed the most prescribed antibiotic type had comparable risks to those prescribed the second most prescribed type, e.g., cHR of 1.02 (95% CI 0.99–1.05) for incident LRTI; however, patients prescribed less frequent types had increased risks of infection-related hospital admission, e.g., cHR of 1.72 (95% CI 1.67–1.78) for incident LRTI (Table 4). No major effect modification in the cHRs were observed when stratifying by sex, age, and period regarding COVID-19 status. cHRs of antibiotic exposure and antibiotic types for incident and prevalent common infections and cHRs of models with stratifications by sex categories, age categories, and period are presented (Table 4 and S13 and S14 Tables in S3 Appendix).

## Further evaluation with pre-pandemic data

For further evaluation of impact of COVID-19 pandemic, we used the pre-pandemic data to develop and validate Cox models for infection-related hospital admissions. See S4 Appendix for more information. S5 Appendix provides the TRIPOD checklist.

## Discussion

The risk prediction models indicated that the main drivers of infection-related hospital admission were age, CCI, and history of prior antibiotics. Antibiotics were found to be more effective in preventing complications (compared to no treatment) in LRTI and UTI, in contrast to URTI. The models found that first-choice antibiotic types were associated with more reduction in the risk of infection-related hospital admission, whereas less popular antibiotic types were associated with lesser effects.

**Table 4. Crude hazard ratios of antibiotic exposure stratified by incident and prevalent common infection (using data from 1 January 2019 to 31 August 2022).**

| | LRTI[1] | | URTI[2] | | UTI[3] | |
| | Crude HR[4] (95% CI[5]) | | Crude HR[4] (95% CI[5]) | | Crude HR[4] (95% CI[5]) | |
| | Incident | Prevalent | Incident | Prevalent | Incident | Prevalent |
|---|---|---|---|---|---|---|
| **Antibiotic exposure** | | | | | | |
| No exposure | reference | reference | reference | reference | reference | reference |
| Exposed | 0.35 (0.35–0.36) | 0.51 (0.48–0.53) | 1.04 (1.03–1.06) | 0.87 (0.84–0.91) | 0.45 (0.44–0.46) | 0.53 (0.51–0.55) |
| **Antibiotic type[6]** | | | | | | |
| Most prescribed | reference | reference | reference | reference | reference | reference |
| Second most prescribed | 1.02 (0.99–1.05) | 0.90 (0.83–0.97) | 1.00 (0.98–1.03) | 0.94 (0.88–1.00) | 1.15 (1.12–1.18) | 0.93 (0.88–1.00) |
| Others | 1.72 (1.67–1.78) | 1.42 (1.32–1.53) | 1.43 (1.40–1.47) | 1.36 (1.28–1.44) | 1.40 (1.36–1.43) | 1.14 (1.08–1.20) |
| None | 3.16 (3.09–3.24) | 2.17 (2.02–2.32) | 1.05 (1.03–1.07) | 1.26 (1.19–1.33) | 2.49 (2.43–2.55) | 1.96 (1.87–2.06) |
| **Stratified by sex** | | | | | | |
| Male | 0.34 (0.33–0.35) | 0.50 (0.47–0.54) | 1.07 (1.05–1.10) | 0.91 (0.86–0.96) | 0.54 (0.52–0.55) | 0.58 (0.54–0.61) |
| Female | 0.36 (0.35–0.37) | 0.51 (0.48–0.54) | 1.00 (0.98–1.03) | 0.86 (0.82–0.90) | 0.39 (0.38–0.40) | 0.51 (0.48–0.53) |
| **Stratified by age** | | | | | | |
| 18–24 | 0.32 (0.26–0.38) | 0.37 (0.24–0.57) | 0.88 (0.82–0.95) | 0.70 (0.59–0.83) | 0.37 (0.33–0.43) | 0.60 (0.45–0.80) |
| 25–34 | 0.32 (0.28–0.36) | 0.55 (0.42–0.73) | 0.86 (0.81–0.92) | 0.68 (0.58–0.79) | 0.43 (0.38–0.48) | 0.45 (0.36–0.56) |
| 35–44 | 0.26 (0.23–0.29) | 0.46 (0.36–0.58) | 0.79 (0.74–0.84) | 0.85 (0.71–1.01) | 0.39 (0.35–0.43) | 0.57 (0.46–0.72) |
| 45–54 | 0.26 (0.24–0.28) | 0.47 (0.39–0.56) | 0.93 (0.88–0.98) | 0.81 (0.71–0.92) | 0.37 (0.34–0.40) | 0.43 (0.36–0.51) |
| 55–64 | 0.27 (0.25–0.28) | 0.49 (0.43–0.56) | 1.02 (0.97–1.06) | 0.93 (0.83–1.03) | 0.44 (0.41–0.47) | 0.49 (0.43–0.56) |
| 65–74 | 0.31 (0.30–0.32) | 0.41 (0.37–0.46) | 1.08 (1.05–1.12) | 0.88 (0.81–0.95) | 0.41 (0.39–0.43) | 0.50 (0.46–0.54) |
| 75+ | 0.42 (0.41–0.43) | 0.55 (0.51–0.58) | 1.08 (1.05–1.10) | 0.89 (0.84–0.94) | 0.48 (0.47–0.50) | 0.57 (0.54–0.60) |
| **Stratified by time** | | | | | | |
| Pre-pandemic | 0.38 (0.36–0.39) | 0.55 (0.52–0.59) | 1.05 (1.03–1.08) | 0.88 (0.83–0.92) | 0.44 (0.42–0.45) | 0.54 (0.51–0.58) |
| Beginning and during pandemic | 0.40 (0.37–0.42) | 0.49 (0.43–0.55) | 1.16 (1.11–1.20) | 1.04 (0.95–1.15) | 0.40 (0.38–0.41) | 0.50 (0.46–0.54) |
| After 2nd lockdown | 0.37 (0.36–0.38) | 0.52 (0.49–0.55) | 1.10 (1.08–1.12) | 0.90 (0.86–0.94) | 0.41 (0.40–0.42) | 0.52 (0.49–0.54) |

[1] LRTI, Lower Respiratory Tract Infection.

[2] URTI, Upper Respiratory Tract Infection.

[3] UTI, Urinary Tract Infection.

[4] HR, hazard ratio.

[5] CI, confidence interval.

[6] The most prescribed and the second most prescribed type of antibiotic are, respectively, amoxicillin and doxycycline for LRTI and URTI, and nitrofurantoin and trimethoprim for UTI.

Prescribed antibiotics were strongly associated with reduced risk of hospital admission related to incident LRTI and UTI, but not in URTI. We could not find much literature on the effectiveness of antibiotics in reducing the risk of hospital admission with common infections. There may be different possible explanations for our finding of greater antibiotic effectiveness in preventing hospital admission related to LRTI and UTI. The first one may be that viral infections may be more frequent with infections, e.g., URTI. General practice in the United Kingdom (UK) does not routinely test whether common infections are bacterial or viral. The second explanation may be due to differential confounding, where antibiotics are prescribed to sicker patients in infections like URTI and healthier patients in infections like LRTI and UTI. However, we did not find substantive evidence for this and a study in a different database found that antibiotic prescribing was unrelated to patient's risk of hospital admission for infection-related complications [3]. The finding in this study of reduced antibiotic effects in preventing complications with increasing count of antibiotic prescription in the one year before is

consistent with a previous study [15]. We also found that antibiotics prescribed for infections like LRTI and UTI were more effective in preventing complications than infections like URTI. The first choice antibiotic types for respiratory infections were amoxicillin and doxycycline and nitrofurantoin and trimethoprim for UTI, which correspond with the recommendations of the National Institute for Health and Care Excellence (NICE) guidelines [16–18]. These first-choice antibiotic types for each infection were associated with a larger reduction of infection-related complication risk, compared with lesser effects for other antibiotic types. The NICE guidelines will have considered the most frequent pathogens in determining the recommended antibiotic type. It could be useful if guidelines also state the antibiotic types that are likely to be ineffective and should not be prescribed.

Infection-related hospital admissions reduced in the beginning and fluctuated during the COVID-19 pandemic, which can be interpreted as the consequence of less transmission of infections and lower incidence of common infections. The latest report on antibiotics utilisation in England stated 10.9% decline in antibiotics consumption between 2019 to 2020, which is an evident impact of COVID-19 pandemic [2]. An analysis of prescribing of first-line antibiotics in English primary care found a 13.5% reduction between March and September 2020 compared with March and September 2019 [19]. A qualitative interview study with GPs in England found that although GPs were more likely to prescribe empirical antibiotics for respiratory tract infections, they prescribed less antibiotics during the pandemic, except for UTI and skin infections [20], which is similar to our findings. International studies reported a similar decrease in antibiotics prescriptions, e.g., in Spain [21], Belgium [22], Brazil [23], and Netherlands [24], in the beginning of the pandemic. The reduction of antibiotics used could have unintended consequences like severe infections and complications [4], which we tried to address in this study.

There are only a limited number of risk prediction models for the prognosis of common infections. Before the COVID-19 pandemic, risk prediction models were developed to predict complications related to incident LRTI, URTI, and UTI in UK primary care [3]. This study's HRs for infection-related hospital admissions are similar to our findings; greater age and CCI increase the risk of complications, but some variables like white ethnicity or winter season have a different association in the current study. Another similar study investigated antibiotics prescribing for common infections in UK primary care, and found age, sex, region, and CCI as associating factors with prescribing antibiotics [25]. Unlike these two studies and our study, others focused on risk prediction models for a specific infection, e.g., UTI [26, 27], sepsis [28], or pneumonia [29], and specific resistance strains, such as carbapenem-resistant Enterobacterales [30, 31].

There are several strength and limitations of this study. The main strength of this study is that it only considered complications related to common infections in patients without COVID-19. Another strength of this study is that it employed a large national EHR dataset which made it possible to develop multiple prediction models for each common infection. This is the first study to look at infection-related hospital admission following prevalent common infections in primary care. This is of importance since there are no guidelines in England for repeated infection, except for UTI. Models for hospital admissions related to prevalent common infections target patients who may be at higher risk of complications. One weakness of our models developed with overall data (from 1 January 2019 to 31 August 2022) is that their accuracy can be challenged since they were developed with the fluctuating records of infections in primary care during the pandemic. Therefore, our main focus was on models with pre-pandemic data. This also highlights the need for updating risk prediction models for infection-related hospital admission in the future, especially those applied in clinical practice. A further limitation relates to the exclusion of patients with COVID-19 given uncertainty

whether all COVID-19 tests performed by private companies were included in the SGSS. Records of infection diagnosis were based on GP consultations, either in person or virtual, which the latter became common during the COVID-19 pandemic and could potentially impact infection diagnosis remotely. The risk prediction models also did not include severity of the infections as signs and symptoms are generally not well coded by GPs. Confounding by severity of infection also was not controlled for in our analyses.

## Conclusion

The COVID-19 pandemic indirectly impacted the antibiotic treatment for common infections, particularly infections like LRTI. Risk models found that age, CCI, and history of prior antibiotics were the main predictors of infection-related hospital admission. Antibiotics appeared more effective in preventing infection-related complications with LRTI and UTI, but not URTI. A focus on risk-based antibiotic prescribing could help to tackle AMR in primary care. There is a need for GPs and patients to be provided with personalised information on the infection risks and prognosis.

## Supporting information

**S1 Appendix. Remaining baseline characteristics of common infections (lower respiratory tract infection, upper respiratory tract infection, and urinary tract infection) and baseline characteristics of the cohort of other infections (sinusitis, otitis media, otitis externa, specific URTI, cough, cold with cough, and sore throat).**
(DOCX)

**S2 Appendix. Remaining counts and rates of common infections (lower respiratory tract infection, upper respiratory tract infection, and urinary tract infection) and counts and rates of the cohort of other infections (sinusitis, otitis media, otitis externa, specific URTI, cough, cold with cough, and sore throat).**
(DOCX)

**S3 Appendix. Performance, adjusted hazard ratios, and crude hazard ratios of Cox models with overall data.**
(DOCX)

**S4 Appendix. Performance, adjusted hazard ratios, and calibrations plots of Cox models with pre-pandemic data.**
(DOCX)

**S5 Appendix. TRIPOD checklist for prediction model development and validation.**
(DOCX)

## Acknowledgments

We are very grateful for all the support received from the TPP Technical Operations team throughout this work, and for generous assistance from the information governance and database teams at NHS England and the NHS England Transformation Directorate. We acknowledge the support from the members of the OpenSAFELY collaborative, namely Ben Goldacre (Director), Brian MacKenna (Director of NHS Service Analytics), Louis Fisher (Data Scientist), Jon Massey (Data Scientist), Amir Mehrkar (Director of Information Governance and External Relations), and Seb Bacon (Chief Technical Officer). The lead author from the

OpenSAFELY collaborative was Brian MacKenna (brian.mackenna@nhs.net). Details of OpenSAFELY Team is available here: https://www.opensafely.org/team/.

## Information governance and ethical approval

NHS England is the data controller of the NHS England OpenSAFELY COVID-19 Service; TPP is the data processor; all study authors using OpenSAFELY have the approval of NHS England [32]. This implementation of OpenSAFELY is hosted within the TPP environment which is accredited to the ISO 27001 information security standard and is NHS IG Toolkit compliant [33].

Patient data has been pseudonymised for analysis and linkage using industry standard cryptographic hashing techniques; all pseudonymised datasets transmitted for linkage onto Open-SAFELY are encrypted; access to the NHS England OpenSAFELY COVID-19 service is via a virtual private network (VPN) connection; the researchers hold contracts with NHS England and only access the platform to initiate database queries and statistical models; all database activity is logged; only aggregate statistical outputs leave the platform environment following best practice for anonymisation of results such as statistical disclosure control for low cell counts [34].

The service adheres to the obligations of the UK General Data Protection Regulation (UK GDPR) and the Data Protection Act 2018. The service previously operated under notices initially issued in February 2020 by the Secretary of State under Regulation 3(4) of the Health Service (Control of Patient Information) Regulations 2002 (COPI Regulations), which required organisations to process confidential patient information for COVID-19 purposes; this set aside the requirement for patient consent [35]. As of 1 July 2023, the Secretary of State has requested that NHS England continue to operate the Service under the COVID-19 Directions 2020 [36]. In some cases of data sharing, the common law duty of confidence is met using, for example, patient consent or support from the Health Research Authority Confidentiality Advisory Group [37].

Taken together, these provide the legal bases to link patient datasets using the service. GP practices, which provide access to the primary care data, are required to share relevant health information to support the public health response to the pandemic, and have been informed of how the service operates. This study was approved by the Health Research Authority and NHS Research Ethics Committee [REC reference 21/SC/0287].

## Author Contributions

**Conceptualization:** Tjeerd Pieter van Staa.

**Data curation:** Ali Fahmi, Victoria Palin, Xiaomin Zhong, Ya-Ting Yang, Jon Massey.

**Formal analysis:** Ali Fahmi, Xiaomin Zhong, Ya-Ting Yang.

**Funding acquisition:** Tjeerd Pieter van Staa.

**Methodology:** Ali Fahmi, Victoria Palin, Tjeerd Pieter van Staa.

**Project administration:** Tjeerd Pieter van Staa.

**Resources:** Simon Watts, Ben Goldacre, Brian MacKenna, Louis Fisher, Jon Massey, Amir Mehrkar, Seb Bacon, Kieran Hand.

**Supervision:** Tjeerd Pieter van Staa.

**Validation:** Ali Fahmi.

**Visualization:** Ali Fahmi.

**Writing – original draft:** Ali Fahmi, Tjeerd Pieter van Staa.

**Writing – review & editing:** Ali Fahmi, Darren M. Ashcroft, Kieran Hand, Tjeerd Pieter van Staa.

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
