## [Decision Letter · Decision Letter 0]

16 Aug 2024

PONE-D-24-06474Evaluation of the impact of COVID-19 pandemic on hospital admission related to common infectionsPLOS ONE

Dear Dr. Fahmi,

Thank you for submitting your manuscript to PLOS ONE. After careful consideration, we feel that it has merit but does not fully meet PLOS ONE’s publication criteria as it currently stands. Therefore, we invite you to submit a revised version of the manuscript that addresses the points raised during the review process.

**ACADEMIC EDITOR: **Upon revision, please incorporate all comments given by the reviewers and editor.Find the PDF attachment and revise it accordingly.==============================

We look forward to receiving your revised manuscript.

Kind regards,

Muluneh Assefa

Academic Editor

PLOS ONE

Journal Requirements:

"This work was supported by Health Data Research UK (Better prescribing in frail elderly people with polypharmacy: learning from practice and nudging prescribers into better practice -BetterRx) and by National Institute for Health and Care Research (NIHR130581 - Cluster randomised trial to improve antibiotic prescribing in primary care: individualised knowledge support during consultation for general practitioners and patients – BRIT2). DMA is funded by the NIHR Greater Manchester Patient Safety Translational Research Centre (PSTRC-2016-003). The views expressed are those of the authors and not necessarily those of Health Data Research UK, the NHS, the NIHR, the Department of Health and Social Care or Public Health England."

"All authors declare the following: BG and OpenSAFELY has received research funding from the Laura and John Arnold Foundation, the NHS National Institute for Health Research (NIHR), the NIHR School of Primary Care Research, NHS England, the NIHR Oxford Biomedical Research Centre, the Mohn-Westlake Foundation, NIHR Applied Research Collaboration Oxford and Thames Valley, the Wellcome Trust, the Good Thinking Foundation, Health Data Research UK, the Health Foundation, the World Health Organisation, UKRI MRC, Asthma UK, the British Lung Foundation, and the Longitudinal Health and Wellbeing strand of the National Core Studies programme; he is a Non-Executive Director at NHS Digital; he also receives personal income from speaking and writing for lay audiences on the misuse of science. AM has received consultancy fees (from https://inductionhealthcare.com) and is member of RCGP health informatics group and the NHS Digital GP data Professional Advisory Group that advises on access to GP Data for Pandemic Planning and Research (GDPPR). For the latter, he received payment for the GDPPR role. "

4. One of the noted authors is a group or consortium "the OpenSAFELY collaborative". In addition to naming the author group, please list the individual authors and affiliations within this group in the acknowledgments section of your manuscript. Please also indicate clearly a lead author for this group along with a contact email address.

**Additional Editor Comments:**

In your revised manuscript, include the following sections:

-Keywords

-Abbreviations

Reviewers' comments:

Reviewer's Responses to Questions

**Comments to the Author**

1. Is the manuscript technically sound, and do the data support the conclusions?

Reviewer #1: Yes

Reviewer #2: Yes

2. Has the statistical analysis been performed appropriately and rigorously? 

Reviewer #1: Yes

Reviewer #2: Yes

3. Have the authors made all data underlying the findings in their manuscript fully available?

Reviewer #1: Yes

Reviewer #2: No

4. Is the manuscript presented in an intelligible fashion and written in standard English?

Reviewer #1: Yes

Reviewer #2: Yes

5. Review Comments to the Author

Reviewer #1: The manuscript is well-written and scientifically sound.The abstract provides a concise overview of the study's background, methods, results, and conclusions, effectively conveying the key findings of the research. The background succinctly outlines the global challenge of antimicrobial resistance and the impact of the COVID-19 pandemic on antibiotic prescribing practices for common bacterial infections, setting the context for the study.

The methods section provides clear details on the data source, inclusion criteria, and statistical analysis used, demonstrating a rigorous approach to studying the risk of hospital admissions related to common infections. The use of electronic health records from The Phoenix Partnership (TPP) through the OpenSAFELY platform, with approval from NHS England, lends credibility to the study's methodology.

The results section presents key findings regarding infection diagnoses, hospital admissions, and the effectiveness of antibiotics in preventing complications. The inclusion of specific counts of hospital admissions for different infections and the comparison between antibiotic-prescribed and non-prescribed groups adds depth to the analysis. The observed reductions in hospital admission risks for lower respiratory tract infection (LRTI) and urinary tract infection (UTI) compared to upper respiratory tract infection (URTI) provide valuable insights into the differential impact of antibiotic prescribing on infection outcomes.

The conclusions drawn from the study findings are logical and supported by the data presented. The emphasis on the large effectiveness of antibiotics in preventing complications related to LRTI and UTI underscores the importance of targeted antibiotic prescribing practices to mitigate risks of hospital admissions for these infections.

Overall, the abstract effectively communicates the study's objectives, methodology, and key findings, contributing valuable insights into the impact of antibiotic prescribing on hospital admissions for common infections during the COVID-19 pandemic. The findings have significant implications for antibiotic stewardship efforts and healthcare decision-making. However, providing additional context on the limitations of the study and potential implications for clinical practice could enhance the abstract's comprehensiveness.

Reviewer #2: It is essential to investigate the impact of the COVID-19 pandemic on the emergence of new antimicrobial resistance. However, certain revisions are required to the submitted manuscript.

Major comments

#1. In lines 31-32, the authors should incorporate the total number of study participants retrieved to conclude the present result.

#2. In the study population section (in lines; 82-96), the authors ought to present clear inclusion and exclusion criteria for study participants in a separate paragraph.

#3. The statistical methods section stated (in lines 104-144) needs to be re-written in a well understandable manner. It is somewhat too long, the authors should focus on major points.

#4. The authors ought to present the result section (in lines; 145-187) in a separate sub-headings like; Baseline characteristics of study participants, proportion and rates of COVID-19-related hospital admission, hazard ratio results, etc.

Minor comments

- Some references are incomplete and missing volume, issue, or page number. Please revise all the reference sections following PLoS ONE referencing guidelines.

6. PLOS authors have the option to publish the peer review history of their article (what does this mean?). If published, this will include your full peer review and any attached files.

Reviewer #1: **Yes: **Doaa Attia

Reviewer #2: No

---

## [Author Response · Author response to Decision Letter 0]

14 Sep 2024

Response: We very much thank the academic editor and reviewers for their detailed and thoughtful comments.

Editor:

Dear Dr. Fahmi, Thank you for submitting your manuscript to PLOS ONE. After careful consideration, we feel that it has merit but does not fully meet PLOS ONE’s publication criteria as it currently stands. Therefore, we invite you to submit a revised version of the manuscript that addresses the points raised during the review process.

ACADEMIC EDITOR: 

• Upon revision, please incorporate all comments given by the reviewers and editor. 

• Find the PDF attachment and revise it accordingly. 

If applicable, we recommend that you deposit your laboratory protocols in protocols.io to enhance the reproducibility of your results. Protocols.io assigns your protocol its own identifier (DOI) so that it 2 can be cited independently in the future. For instructions see: https://journals.plos.org/plosone/s/submission-guidelines#loc-laboratory-protocols [c05y1x9s.r.useast-2.awstrack.me]. Additionally, PLOS ONE offers an option for publishing peer-reviewed Lab Protocol articles, which describe protocols hosted on protocols.io. Read more information on sharing protocols at https://plos.org/protocols?utm_medium=editorialemail&utm_source=authorletters&utm_campaign=protocols [c05y1x9s.r.us-east-2.awstrack.me]. We look forward to receiving your revised manuscript. 

Kind regards, 

Muluneh Assefa 

Academic Editor 

PLOS ONE

Comments in the PDF file

modify your topic by incorporating the context "antimicrobial resistance". Because you have mentioned about the impact of COVID-19 on antimicrobial resistance and hospital admission in your abstract and introduction.

Response: We modified the title as follows: “Evaluation of the impact of COVID-19 pandemic on hospital admission related to common infections: risk prediction models to tackle antimicrobial resistance in primary care”.

Rewrite your objective

Response: We have rewritten the objective as follows: “The objectives of this study were to evaluate the impact of the COVID-19 pandemic on treatment of common infections, develop risk prediction models and examine the effects of antibiotics on infection-related hospital admissions”.

Correct your grammar

Response: The manuscript has been reviewed by a scientific writer. Several edits were made, and some text also deleted. Please note that some sections around OpenSAFELY policies are mandatory text (as outlined in this webpage: https://www.opensafely.org/policies-for-researchers/).

The conclusion should answer your topic. please revise it and add strong recommendations

Response: We added the following stronger recommendations to the conclusion: “A substantial variation in hospital admission risks between infections and patient groups was found. Antibiotics appeared more effective in preventing infection-related complications with LRTI and UTI, but not URTI. While this study has several limitations, the results indicate that a focus on risk-based antibiotic prescribing could help tackle AMR in primary care.”

Why adults only?. You should rationalized in your introduction section.

Response: We have added the following sentence to Study population subsection: “The reason for restricting to adults was that the prevalence of common infections is different in children and that the BRIT2 project is focusing on adults [7]”.

Give some information about the study design and cite Figure 1. Like this..........................(Figure 1).

Response: We cited the figure as suggested and added further information on the study design as follows: “The diagram of study design shows the inclusion criterion of any infection-related hospital admission and the exclusion criterion of having any COVID-19 diagnosis records (Fig 1)”.

Study variables

Response: Done.

Cite tables and figures like this

Response: We cited all tables and figures as suggested.

-Discuss your findings briefly by comparing with another study conducted elsewhere.

-Be smart and write to the point.

Response: We edited the Discussion (and other parts of the manuscript).

such as

Response: Done.

reference

Response: Reference is our own findings in the results section.

Rewrite this section according to journal requirement.

Response: Done.

write full form at the bottom as legend

Response: Done.

Journal Requirements

1. Please ensure that your manuscript meets PLOS ONE's style requirements, including those for file naming. The PLOS ONE style templates can be found at https://journals.plos.org/plosone/s/file?id=wjVg/PLOSOne_formatting_sample_main_body.pdf [c05y1x9s.r.us-east-2.awstrack.me] and https://journals.plos.org/plosone/s/file?id=ba62/PLOSOne_formatting_sample_title_authors_affiliat ions.pdf [c05y1x9s.r.us-east-2.awstrack.me] 

Response: We updated the manuscript according to PLOS ONE’s style requirements. 

2. Please state what role the funders took in the study. If the funders had no role, please state: ""The funders had no role in study design, data collection and analysis, decision to publish, or preparation of the manuscript."" If this statement is not correct you must amend it as needed. Please include this amended Role of Funder statement in your cover letter; we will change the online submission form on your behalf.

Response: We updated the Funding section and added the above statement since the funders had no role. The updated part of the funding is as follows: “AF is funded by the National Institute for Health and Care Research (DSE Award; NIHR303781). The funders had no role in study design, data collection and analysis, decision to publish, or preparation of the manuscript”.

3. Thank you for stating the following in the Competing Interests section. Please confirm that this does not alter your adherence to all PLOS ONE policies on sharing data and materials, by including the following statement: ""This does not alter our adherence to PLOS ONE policies on sharing data and materials.” (as detailed online in our guide for authors http://journals.plos.org/plosone/s/competing-interests). If there are restrictions on sharing of data and/or materials, please state these. Please note that we cannot proceed with consideration of your article until this information has been declared.

Response: We added a sentence to the Conflict of Interest section about the restrictions on the access to the data of the study, referring to the Data access and verification section that is written based on OpenSAFELY’s Policies for Researchers (https://www.opensafely.org/policies-for-researchers/).

4. One of the noted authors is a group or consortium "the OpenSAFELY collaborative". In addition to naming the author group, please list the individual authors and affiliations within this group in the acknowledgments section of your manuscript. Please also indicate clearly a lead author for this group along with a contact email address.

Response: We updated the Acknowledgements section by adding the list of individual authors, their affiliations, and the name and email address of the lead author from the OpenSAFELY collaborative.

Response: We added the full name of the ethics committee that approved our study, the reference number, and the reason why consent was not needed. It reads as follows: “Our study protocol was approved by the Research Ethics Committee on 17 August 2021 (reference 21/SC/0287). Patients’ consent was not required since the data were anonymised electronic health records (EHRs) and for retrospective research use”.

Response: We added the captions of Supporting Information files to the end of the manuscript.

Response: We updated the list of references.

8. In your revised manuscript, include the following sections: Keywords and Abbreviations.

Response: We added the keywords and abbreviations sections after the abstract.

Reviewer #1:

The manuscript is well-written and scientifically sound. The abstract provides a concise overview of the study's background, methods, results, and conclusions, effectively conveying the key findings of the research. The background succinctly outlines the global challenge of antimicrobial resistance and the impact of the COVID-19 pandemic on antibiotic prescribing practices for common bacterial infections, setting the context for the study.

The methods section provides clear details on the data source, inclusion criteria, and statistical analysis used, demonstrating a rigorous approach to studying the risk of hospital admissions related to common infections. The use of electronic health records from The Phoenix Partnership (TPP) through the OpenSAFELY platform, with approval from NHS England, lends credibility to the study's methodology. 

The results section presents key findings regarding infection diagnoses, hospital admissions, and the effectiveness of antibiotics in preventing complications. The inclusion of specific counts of hospital admissions for different infections and the comparison between antibiotic-prescribed and nonprescribed groups adds depth to the analysis. The observed reductions in hospital admission risks for lower respiratory tract infection (LRTI) and urinary tract infection (UTI) compared to upper respiratory tract infection (URTI) provide valuable insights into the differential impact of antibiotic prescribing on infection outcomes. 

The conclusions drawn from the study findings are logical and supported by the data presented. The emphasis on the large effectiveness of antibiotics in preventing complications related to LRTI and UTI underscores the importance of targeted antibiotic prescribing practices to mitigate risks of hospital admissions for these infections. 

Overall, the abstract effectively communicates the study's objectives, methodology, and key findings, contributing valuable insights into the impact of antibiotic prescribing on hospital admissions for common infections during the COVID-19 pandemic. The findings have significant implications for antibiotic stewardship efforts and healthcare decision-making. However, providing additional context on the limitations of the study and potential implications for clinical practice could enhance the abstract's comprehensiveness.

Response: We updated the abstract and highlighted limitations. The conclusion now reads: “While this study has several limitations, the results indicate that a focus on risk-based antibiotic prescribing could help tackle AMR in primary care”.

Reviewer #2:

It is essential to investigate the impact of the COVID-19 pandemic on the emergence of new antimicrobial resistance. However, certain revisions are required to the submitted manuscript. Major comments

1. In lines 31-32, the authors should incorporate the total number of study participants retrieved to conclude the present result. 

Response: The total number of study participants are incorporated in the Results section of the Abstract; therefore, we did not incorporate them in lines 31-32 which are the beginning of the Methods section of the Abstract. Please note that some sections of the manuscript, including the Methods section of the Abstract, must follow OpenSAFELY’s policies for researchers: https://www.opensafely.org/policies-for-researchers/.

2. In the study population section (in lines; 82-96), the authors ought to present clear inclusion and exclusion criteria for study participants in a separate paragraph. 

Response: We re-wrote the Study population subsection and presented the following paragraph with clear inclusion and exclusion criteria: “The inclusion criteria were adult individuals registered with a general practice during study period (1 January 2019 to 31 August 2022), who had diagnosis of a common infection, namely LRTI, URTI (including specific URTI, cough, cold with cough, and sore throat), lower UTI (not including renal infections), sinusitis, otitis media, and otitis externa. The reason for restricting to adults was that the prevalence of common infections is different in children and that the BRIT2 project is focusing on adults [7]. Individuals with a record of COVID-19 diagnosis in the 90 days before or 30 days after the date of infection diagnosis were excluded. This was based on data from the Second Generation Surveillance System (SGSS) with positive COVID-19 tests and those of the general practitioners (GPs) with a COVID-19 diagnosis. The diagram of study design shows the inclusion criterion of any infection-related hospital admission and the exclusion criterion of having any COVID-19 diagnosis records (Fig 1)”.

3. The statistical methods section stated (in lines 104-144) needs to be re-written in a well understandable manner. It is somewhat too long, the authors should focus on major points. 

Response: We have rewritten the Statistical methods subsection and substantially reduced it.

4. The authors ought to present the result section (in lines; 145-187) in a separate sub-headings like; Baseline characteristics of study participants, proportion and rates of COVID-19-related hospital 6 admission, hazard ratio results, etc. 

Response: We added sub-headings to the results section.

Minor comments

Some references are incomplete and missing volume, issue, or page number. Please revise all the reference sections following PLoS ONE referencing guidelines.

Response: We checked the references and updated the incomplete ones.

---

## [Decision Letter · Decision Letter 1]

20 Sep 2024

Evaluation of the impact of COVID-19 pandemic on hospital admission related to common infections: risk prediction models to tackle antimicrobial resistance in primary care

PONE-D-24-06474R1

Dear Dr. Fahmi,

We’re pleased to inform you that your manuscript has been judged scientifically suitable for publication and will be formally accepted for publication once it meets all outstanding technical requirements.

Kind regards,

Muluneh Assefa

Academic Editor

PLOS ONE

Additional Editor Comments (optional):

Reviewers' comments:

Reviewer's Responses to Questions

**Comments to the Author**

1. If the authors have adequately addressed your comments raised in a previous round of review and you feel that this manuscript is now acceptable for publication, you may indicate that here to bypass the “Comments to the Author” section, enter your conflict of interest statement in the “Confidential to Editor” section, and submit your "Accept" recommendation.

Reviewer #2: All comments have been addressed

2. Is the manuscript technically sound, and do the data support the conclusions?

Reviewer #2: Yes

3. Has the statistical analysis been performed appropriately and rigorously? 

Reviewer #2: Yes

4. Have the authors made all data underlying the findings in their manuscript fully available?

Reviewer #2: Yes

5. Is the manuscript presented in an intelligible fashion and written in standard English?

Reviewer #2: Yes

6. Review Comments to the Author

Reviewer #2: (No Response)

7. PLOS authors have the option to publish the peer review history of their article (what does this mean?). If published, this will include your full peer review and any attached files.

Reviewer #2: No

---

## [Editor Report · Acceptance letter]

18 Oct 2024

PONE-D-24-06474R1 

PLOS ONE

Dear Dr. Fahmi, 

I'm pleased to inform you that your manuscript has been deemed suitable for publication in PLOS ONE. Congratulations! Your manuscript is now being handed over to our production team.

Kind regards, 

on behalf of

Dr. Muluneh Assefa 

Academic Editor

PLOS ONE